# Early Life Oxidative Stress and Long-Lasting Cardiovascular Effects on Offspring Conceived by Assisted Reproductive Technologies: A Review

**DOI:** 10.3390/ijms21155175

**Published:** 2020-07-22

**Authors:** Huixia Yang, Christina Kuhn, Thomas Kolben, Zhi Ma, Peng Lin, Sven Mahner, Udo Jeschke, Viktoria von Schönfeldt

**Affiliations:** 1Department of Obstetrics and Gynecology, University Hospital, LMU Munich, 81377 Munich, Germany; huixia.yang@med.uni-muenchen.de (H.Y.); christina.kuhn@med.uni-muenchen.de (C.K.); thomas.kolben@med.uni-muenchen.de (T.K.); zhi.ma@med.uni-muenchen.de (Z.M.); lin.peng@med.uni-muenchen.de (P.L.); sven.mahner@med.uni-muenchen.de (S.M.); viktoria.schoenfeldt@med.uni-muenchen.de (V.v.S.); 2Department of Obstetrics and Gynecology, University Hospital Augsburg, 86156 Augsburg, Germany

**Keywords:** oxidative stress, long-lasting, cardiovascular, assisted reproductive technologies, offspring

## Abstract

Assisted reproductive technology (ART) has rapidly developed and is now widely practised worldwide. Both the characteristics of ART (handling gametes/embryos in vitro) and the infertility backgrounds of ART parents (such as infertility diseases and unfavourable lifestyles or diets) could cause increased oxidative stress (OS) that may exert adverse influences on gametogenesis, fertilisation, and foetation, even causing a long-lasting influence on the offspring. For these reasons, the safety of ART needs to be closely examined. In this review, from an ART safety standpoint, the origins of OS are reviewed, and the long-lasting cardiovascular effects and potential mechanisms of OS on the offspring are discussed.

## 1. Introduction

The use of assisted reproductive technology (ART) began in 1978. Since then, ART has been widely used worldwide [1]. More than eight million human babies are estimated to have been conceived through ART [2], and the annual increase in this number is estimated to average at 9.1% per year [1]. ART is developing rapidly, and now includes techniques such as intra-uterine insemination (IUI), in vitro fertilisation (IVF), intracytoplasmic sperm injection (ICSI)), accompanied by controlled ovarian hyperstimulation (COH), oocyte retrieval, embryo culture, and embryo transfer, with or without pre-implantation genetic diagnosis or screening (PGD or PGS), gametes or embryos freezing and thawing, surgical sperm retrieval (SSR), and assisted hatching [3,4].

Despite all advances, it is unavoidable that gametes or embryos are handled in vitro. Because ART occurs at the preimplantation period when gametes or embryos are highly sensitive and experience developmental plasticity, the environmental stimuli may alter the embryonic developmental trajectory. According to the ’developmental origins of adult disease’ (DOHaD) hypothesis [5], environmental exposures in early life can exert a long-lasting influence on health and lead to adult-onset chronic non-communicable diseases (NCDs) such as hypertension [6], cardiovascular diseases (CVDs) [7], and type 2 diabetes mellitus (T2DM) [8]. ART can be regarded as an extreme ’exposure’, despite the fact that conclusions of ART safety studies are conflicting, the possible adverse effects have been linked to birth defects [9], epigenetic diseases [10], dysfunction of various body systems (e.g., cardiovascular, metabolic, and neurological systems) [11], and paediatric neoplasms (e.g., leukaemia and Hodgkin’s lymphoma) [12,13]. The observed long-term outcomes of ART, including cardiometabolic and neurological NCDs, are consistent with the DOHaD model [14].

Oxidative stress (OS) is related to an excess of reactive oxygen species (ROS) and a decrease in antioxidant enzymes. This concept was established by Helmut in 1985 [15]. Excess ROS has been proposed to cause severe damage during embryonic development [16], especially in the cardiovascular system, because it is one of the first functional systems to develop. In the ART area, studies on OS have focused on infertile men and their sperm [17,18]. It is now well established that the main cause of male infertility is OS, which can damage sperm DNA, influencing the health of the offspring [19]. Studies have also revealed that OS plays a vital role in ART outcomes. A study found that a reduction in OS improved ART outcomes [20]. Various factors associated with ROS production in an ART setting have been investigated, and the antioxidant strategy in an ART setting has also been explored [20,21]. Several animal models have been applied to study the effect and associated mechanisms of OS on ART offspring [22]. Nevertheless, to the best of our knowledge, no human epidemiological studies have looked at the effect of OS on ART offspring. On the one hand, the health outcome-related follow-up information of ART offspring in the available databases are insufficiently detailed and are even lacking. On the other hand, no databases have recorded the OS status in an ART setting for ART offspring. In fact, in other medical areas, there have been countless studies on OS/ROS, most of which are focused on the negative effects of excessive ROS/OS. Excessive ROS/OS has been implicated in over 100 diseases [23] (e.g., diabetes mellitus [24], CVDs [25], and neurodegenerative diseases [26]), also playing an important role in the pathogenicity of ageing [27].

In this review, from the ART point of view, we describe the origins of OS, provide a timely synthesis of the current evidence on the long-lasting cardiovascular effects of ART-associated OS, and discuss the potential underlying mechanisms. We expect that our review will inform future OS-associated research in the ART area as well as propose suggestions for preventing adverse health outcomes in ART offspring.

## 2. Origins of OS

### 2.1. Paternally Derived OS

OS has been linked to a variety of male fertility complications, including leukocytospermia [28], varicocele [29], cryptorchidism [30], spermatic cord torsion [31], male accessory gland infections (MAGI) [32], advanced age [33], obesity [34], diabetes [35], and autoimmune disorders [33]. Infertile men are more likely to possess excessive levels of ROS compared to fertile men [36], which has been identified as one of the few defined aetiologies for male infertility [37]. Recently, male oxidative stress infertility (MOSI) has been proposed to describe infertile men with OS and abnormal semen characteristics. This term includes many patients previously classified as having male idiopathic infertility [38].

In the male genital tract, in addition to the ROS generated from sperm cells [39], other cells may also produce ROS. Among them, leukocytes can produce ROS at levels 1000 times higher than that of sperm at capacitation [40] and may contribute to OS [41]. Further compounding this issue, the plasma membrane of sperm cells contains large quantities of polyunsaturated fatty acids (PUFAs), making them particularly susceptible to elevated ROS levels during OS [42]. OS can also negatively influence other sperm components (i.e., nucleic acids and proteins), inducing sperm DNA fragmentation (SDF) and low sperm motility [43]. Furthermore, compared with somatic cells, there is a lack of cytoplasm and poorer antioxidant capacity in mature spermatozoa, thereby rendering it more vulnerable to OS [44]. Nevertheless, it is entirely possible for sperm suffering from oxidative DNA damage to fertilise an oocyte and thus possibly exert adverse effects on the offspring [45], especially in the context of ICSI [46]. On the other hand, lifestyle and diet factors such as cigarette smoking [47], alcohol abuse [48], psychological stress [49], recreational and illicit drugs use [17,50], malnutrition [51], and excessive physical activity [52]; environmental and occupational exposures such as air pollution [53], radiation [54], heat stress [55], plasticizers (e.g., phthalates) [56], heavy metals (e.g., cadmium) [57], and pesticide/herbicides [33]; and special treatments such as radiation therapy and chemotherapy have been linked with OS [58,59] (as shown in Figure 1).

### 2.2. Maternally Derived OS

Compared with studies of OS and male sperm, it appears that fewer studies have focused on OS and oocytes/oocyte-cumulus complexes. Nevertheless, it cannot be assumed that this point is less important in oogenesis, fertilisation, pregnancy, and production of healthy offspring. After all, the earliest determinant of life potential is the oocyte. In the female reproductive system, the uterine environment, fallopian tubes, and follicular fluid are the main sources generating ROS [60,61,62]. Normal levels of ROS are responsible for pregnancy establishment in IVF cycles, while excess ROS in the follicular fluid can present a substantial threat to successful assisted reproduction [63].

Recent studies have also shown that OS may cause absence of the oocyte meiotic spindle and may be closely associated with low fertilization rates, compromised embryonic quality, and decreased clinical pregnancy rates [64]. In fact, women attending ART units are usually of advanced age and/or have been diagnosed with other diseases (e.g., endometriosis [65], polycystic ovary syndrome (PCOS) [66], hydrosalpinx [67], and obesity [52]). After pregnancy, women who underwent ART have been reported to be affected by higher incidences of several pregnancy complications (e.g., hypertensive disorders of pregnancy (HDP) [68], gestational diabetes mellitus (GDM) [68], intrauterine growth restriction (IUGR) [69], and preterm birth [68]). All these diseases and pregnancy complications are associated with increased OS, which might exert an influence on the offspring’s development [70]. Specifically, during the maternal ageing process, significantly increased OS occurs in the ovarian and follicular environment, causing impaired oocyte quality and compromised oocyte meiosis [71]. Similar to men, unfavourable lifestyles and diets, adverse environmental and occupational exposures, and special treatments can also contribute to excessive OS in women [52,70,72,73,74,75,76,77,78] (as shown in Figure 2). Different from men, women exert OS on the offspring, not only through the fertilised oocytes, but also through the uterine environment throughout the whole pregnancy.

### 2.3. ART-Derived OS

ART requires in vitro manipulations of gametes or embryos in a synthetic culture environment. Due to lack of a natural antioxidant system and factors driving ROS production (Figure 3), it is difficult to maintain pro-oxidant/antioxidant balance in vitro, and the resulting increased OS may have an adverse impact on the embryo/offspring. There are various stimuli of OS in the ART setting, including cryopreservation [79,80], gamete or embryo manipulation [81], visible light [82], pH fluctuations [83], temperature fluctuations [84], fluctuating oxygen tension (Pa, O_2_) [85], centrifugation [86], culture media (especially those containing specific substances, e.g., Fe^2+^ and Cu^2+^) [62], and others. For example, in the oviduct and uterus, under certain physiological conditions, gametes or embryos are exposed to O_2_ concentrations of 2–8 % [87]. During in vitro manipulations, gametes and embryos have a chance of being exposed to higher O_2_ concentrations (e.g., atmospheric O_2_ concentrations around 20–21%). The presence of high concentrations of O_2_ during the incubation stage can activate a variety of cellular oxidase enzymes. This in turn generates excessive ROS, leading to OS [88]. The excess ROS can impact the biological processes of early embryonic development with potentially long-lasting health effects for the offspring.

Collectively, compared with naturally conceived offspring, ART offspring appear to be more likely to suffer from excessive OS. Meanwhile, it should be noted that most of the OS originating from ART parents (e.g., adverse lifestyles and environmental exposures) and ART per se are preventable. For ART-associated OS, the potential management includes oral antioxidant supplements for ART parents [89] and modifications in ART protocols. These include antioxidant supplements to ART culture media [20,89], antioxidant techniques in semen preparation, reduced oocyte-handling time, and minimal exposure of zygotes to atmospheric oxygen concentrations [20]. The ART-associated antioxidants consist of enzymatic antioxidants (e.g., superoxide dismutase, catalase, and the glutathione system), non-enzymatic antioxidants (e.g., Vitamins E, C, and B9 (folic acid), melatonin, coenzyme Q10, and L-carnitine) and combined antioxidants (e.g., Vitamin E + Vitamin C) [89]. Diets containing antioxidant molecules for ART parents may also provide antioxidant benefits [90].

## 3. OS-Associated Mechanisms in ART

### 3.1. Formation of OS

Various stress conditions may contribute to OS with increased production of ROS. ROS (e.g., superoxide (O_2_^•−^), hydroperoxyl (HO_2_^•^), hydroxyl (OH^•^), and peroxyl radicals (RO_2_^•^), and hydrogen peroxide (H_2_O_2_) [91]) are generated from the partial reduction of O_2_ to O_2_^•−^, which occurs as a result of oxygen’s preferential acceptance of one electron at the time of redox reactions [92]. ROS are highly active molecules that are continuously generated by mitochondrial electron transport and enzymes (e.g., nicotinamide adenine dinucleotide phosphate (NADPH)-oxidase, xanthine oxidase, and lipoxygenase) [93]. ROS that originate intracellularly can be released extracellularly [94], and play vital roles in modulating the signaling pathways in response to intra- and extra-cellular stimuli [95]. Mitochondria are the primary source of ROS, resulting from its role in energy (i.e., ATP) production via oxidative phosphorylation (OXPHOS) [92]. The major sites of ROS emission in the respiratory chain are complex I and complex III [96]. During IVF, selected spermatozoa and oocytes are combined in a petri dish with the fertilisation medium and are checked for fertilization after several hours’ incubation. During this time, ROS can be generated from the oocyte/cumulus cell mass and the spermatozoa, due to the cells’ own metabolism, and the levels of ROS production can be elevated due to the lack of a natural antioxidant defence system and various stimuli. Furthermore, during centrifugation, excess ROS can be derived from the spermatozoa because of the absence of antioxidant-rich seminal plasma and the activation of ROS production [20]. In addition, the external environment that surrounds the cells in an ART setting can also induce OS. For example, even though the composition of the commercial culture media changes over time with various suppliers, most contain serum or serum synthetic replacements, vitamins, albumin, and other components (e.g., heavy metal chelators or buffer). Therefore, the medium itself can be a trigger of OS [97]. Other environmental factors can also contribute to OS as mentioned. In response to various environmental stimuli, cells produce certain mediators and intermediates (mostly ROS) to propagate environmental signals to the cell nucleus, affecting gene regulation and transcription while inducing various phenotypic responses (in the form of inflammation and pathogenesis) [98].

### 3.2. Epigenetic Modifications Resulting from OS

Epigenetic modifications refer to dynamic and heritable changes in gene expression without DNA sequence changes. These are profoundly involved in OS responses [99] and are regarded as potential mechanisms that influence the developmental origins of CVDs later in adulthood [100]. Maximal epigenetic reprogramming, characterized as ’dynamic’, ’extremely sensitive’, and ’plastic’, occurs during the early stages of life, coinciding with the time that ART procedures take place [101,102]. Both animal studies and follow-up studies of ART children suggest that ART can cause epigenetic perturbation in offspring [10,103], even at the two-cell stage of embryos [104]. It has been proposed that OS during pregnancy may affect the intrauterine foetus and cause cardiovascular dysfunction in later life through epigenetic modifications [105]. Based on these findings, we speculated that ART-associated OS may also influence the offspring through an epigenetic mechanism. In general, ROS can affect epigenetic modifications through both direct and indirect means [106]. For example, OH^•^ can directly lead to the transformation from 5-methylcytosine (5-mC, a form of DNA methylation) to 5-hydroxymethylcytosine (5-hmC, an intermediate in active DNA demethylation [107]) [108], which has been suggested to interfere with DNA methyltransferase 1 (DNMT1), preventing the proper inheritance of methylation patterns, thereby causing indirect CpG sites demethylation [109]. ROS can also indirectly affect epigenetic modifications. For example, H_2_O_2_-induced OS can impair histone demethylase activity, causing increased global histone methylation of histone H3 lysine 4 (H3K4), histone H3 lysine 27 (H3K27), and histone H3 lysine 9 (H3K9), while H3K4 trimethylation (H3K4me3) appears to be affected most by OS; global acetylation levels show temporary decreases in response to OS and return to normal levels after long-term OS. The activity of DNA demethylases (ten-eleven-translocation (TET) proteins) can also be compromised by OS, inducing global 5-mC increases and 5-hmC decreases [110]. These epigenetic modifications can then regulate gene expression via changes in chromatin accessibility in response to OS [111]. Kietzmann et al. described an ROS-related epigenetic landscape in cardiovascular systems; we direct interested readers to a detailed review [106]. Furthermore, evidence also suggests an interplay between OS and epidemic modifications [112,113]. For example, the down-regulation of SUV39H1 (a H3K9 histone methyltransferase) facilitates the recruitment of SRC-1 (a histone acetyltransferase) and JMJD2C (also known as KDM4C, a H3K9 histone demethylase) with reduced di/trimethylation and acetylation of H3K9 on the promoter of p66Shc (a key driver of mitochondrial OS and vascular damage [114]), which may ultimately drive OS [115].

### 3.3. Nrf2-Mediated Anti-OS Signaling Pathway

The redox-sensitive transcriptional factor nuclear factor erythroid 2-related factor 2 (Nrf2, also known as NFE2L2) is a well-characterized ’master regulator’ of antioxidant gene expression via its activation of the Nrf2-antioxidant response element (ARE) pathway [116]. Nrf2 dysregulation has been implicated in different aspects of CVDs [117] and multiple types of cancers (e.g., ovarian cancer [118], breast cancer [119], and glioblastoma [120]), while Nrf2 itself has been identified as a promising therapeutic target for these chronic diseases resulting from its role in providing cytoprotection against diverse stress and pathologies [121]. Recent preclinical data have revealed that N-palmitoylethanolamine-oxazoline (PEA-OXA), an antioxidant compound, protects against cardiovascular complication through upregulation of Nrf2 and Nrf2-target genes [122]. Other compounds (e.g., linarin (LIN) [123] and resveratrol (RES) [124]) also provide beneficial effects in myocardial ischemia/reperfusion injury and CVDs by activating Nrf2. In utero, excessive OS can trigger a cascade of molecular events, imperilling the health of offspring [125]. The Nrf2-mediated OS response is one of the most important cytoprotective mechanisms against OS as it transcribes many antioxidative genes and ROS-scavenging proteins [126] that are not only closely associated with embryo survival in in vitro conditions [127], foetal development in utero, and cardiometabolic health in childhood or later life [128], but also involved in maintaining vascular homeostasis [129]. Given this evidence, it is reasonable to assume that Nrf2-mediated antioxidant signaling pathway may also serve as an important mechanism for OS-induced cardiovascular effects in ART offspring. Under redox homeostasis, Nrf2 is bound to its inhibitor, Kelch-like ECH-associated protein 1 (Keap1), and is located in the cytoplasm, where it facilitates the ubiquitin-mediated degradation of Nrf2. However, during OS, Nrf2 is phosphorylated and released from Keap1 and is translocated and accumulated in the nucleus, where it heterodimerizes with small musculoaponeurotic fibrosarcoma (Maf) proteins, binds to ARE, and transcriptionally upregulates antioxidant gene expression [130]. The representative antioxidant genes regulated by Nrf2 are summarized in Table 1.

Collectively, in the ART area, increased OS is one of the major triggers of early life genetic/epigenetic changes in the offspring. From a preventive point of view, the adverse influence of OS can possibly be reversed through timely appropriate interventions (e.g., medical supplements for ART children and ART parents, chemical modification of ART culture media), opening a window for the potential prevention of adverse long-lasting effects on ART offspring.

## 4. Long-Lasting Cardiovascular Effects 

In the human body, ROS act as a ’double-edged sword’, playing a paradoxical role. Normal levels of ROS are important regulators of various transcription factors and signal transduction pathways. Excessive ROS levels can lead to OS, causing damage to cellular components (e.g., proteins, lipids, and DNA), mitochondrial dysfunctions, inhibition of oocyte maturation, delayed embryonic development, and induction of apoptosis in embryos [149]. Several lines of evidence suggest that OS plays a key role in the foetal programming of adulthood CVDs [150,151,152,153]. Studies on mammalian offspring suffering from OS (e.g., hypoxia-reoxygenation) during the gestational period reported that these offspring developed endothelial dysfunction [150,151], enhanced myocardial contractility [151], and hypertension [150,152] in adulthood. Foetal programming-induced alterations are transmissible not only throughout life but also in the subsequent generation [22].

From the maternal point of view, advanced maternal age (AMA) [154,155] and OS-increased ART-associated pregnancy complications (e.g., HDP [156,157], GDM [158], IUGR [159], and preterm birth [100]) are frequently associated with cardiovascular dysfunction in the offspring. For example, AMA/HDP/IUGR have been linked with increased blood pressure and/or altered cardiovascular function in offspring [154,155,156,159], while GDM/preterm birth have been linked with CVDs in offspring [100,158]. Specifically, in a mouse model, it was reported that AMA affects the phenotype of the offspring in a sex-dependent manner: in young adulthood (four months of age), male (but not female) offspring birthed by aged dams presented reperfusion injury and impaired endothelium-dependent relaxation. In mature adulthood (12 months of age), female offspring showed increased systolic blood pressure, whereas male offspring showed decreased ventricular diastolic function and increased vascular sensitivity to methacholine [154,155].

Because of the relative novelty of ART, the follow-up times of relevant studies remain limited, and the debate as to whether the ART techniques cause long-lasting adverse effects on the offspring remains ongoing. Epidemiological studies on ART children and young adults revealed that the ART offspring presented cardiovascular problems (e.g. congenital heart defect [160] and postnatal dysfunctions of the cardiovascular system [161,162,163,164]), despite the fact that one study reported, among 22–35-year-old adults, ART did not correlate with an increase in prevalent cardiovascular risk factors. However, the study population in this study was still in early adulthood and the authors only used non-invasive methods to detect early markers of sub-clinical atherosclerosis; therefore, they did not evaluate the relationships with clinical cardiovascular events (e.g., CVDs) [165]. Several systematic reviews and meta-analyses [166,167,168] have been performed on the cardiovascular profiles of offspring conceived by ART, with the results revealing mild but statistically significant cardiovascular differences in ART offspring. As Guo et al. concluded, ART-conceived children showed mildly but statistically significantly elevated blood pressure with sub-optimal diastolic function, thicker blood vessels, and lower levels of low-density lipoprotein cholesterol (LDL-C) [167].

Despite the limited data on human ART offspring, subclinical cardiometabolic alterations are detectable. Nevertheless, because CVDs are chronic, adult-onset diseases and significant signs of CVDs require years to develop. ART is relatively new; therefore, the follow-up time of epidemiological studies on ART remains limited (i.e., the first ’test-tube baby’, Louise Brown, was born in 1978 [169] and was only 42 years old as of 2020). Therefore, it is too early to form a definite conclusion. More long-term (even life-long) follow-up periods are warranted.

## 5. Limitations and Prospects of Current Studies

Regarding human ART studies such as those studying the influence of OS in a cohort, it is not clear whether ART is the culprit for OS or whether the infertility factors of ART parents play confounding roles. Moreover, the spectrum of ART-offspring demographic confounders (e.g., lifestyles, dietary habits, familial socio-economic status, adverse childhood experiences, as well as other life experiences), rapidly changing ART protocols, and the various types of culture media make such research more complex. To guarantee a detailed multivariate examination of this kind of study, a large sample size may be key to ensuring adjustments for various confounders. A comprehensive record of clinical and laboratory parameters (independent variables) and a prospective longitudinal design are also necessary.

To independently study the influence of ART on offspring, the ART mouse is an excellent model (i.e., there is no background of infertility; pregnancy and juvenile periods are short; and it is possible to appropriately replicate complications in humans [168]). Human embryonic stem cells (hESC) can also serve as a novel in vitro model to study the effects of OS on the early embryo [170].

In fact, in human assisted reproduction, both the ART procedure and the infertility backgrounds of ART parents may cause increased OS. Increased OS not only is the reason for patients visiting ART clinics, but it is also the outcome of in vitro ART manipulations. Despite increased OS being a prevalent phenomenon in ART, (i) there are no established uniform indicators of OS or standardized cut-off values for ART, ART men, and ART women; (ii) some ART laboratories make no attempt to test for the presence of OS; (iii) neither do the majority of ART clinics analyse their patients’ OS statuses nor pay attention to the clinical causes and sentinel signs of OS, nor do they suggest any changes to OS-related adverse lifestyles or instigate any measures to maintain the redox balance in their patients. Nevertheless, these approaches will not only increase a couple’s chances of natural conception, but will also optimise the efficiency of the ART and the health of ART offspring.

## 6. Conclusions

OS, which have potentially adverse effects on ART offspring, may derive not only from ART per se but also from the infertility backgrounds of ART parents. OS might exert a long-lasting influences on the offspring’s cardiovascular system via epigenetic and genetic alterations (Figure 4).

## Figures and Tables

**Figure 1 ijms-21-05175-f001:**
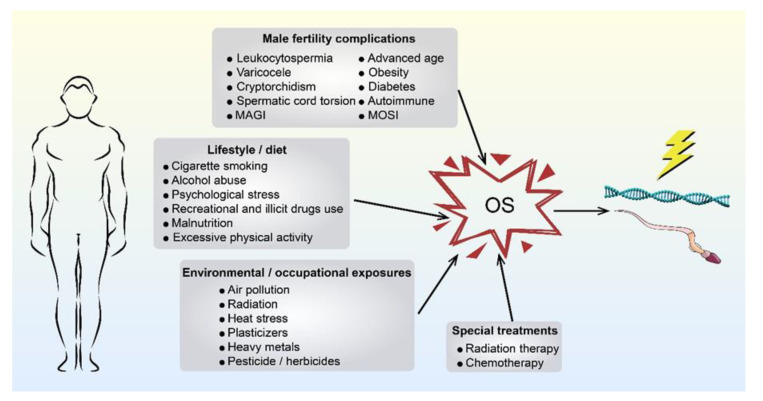
This figure shows the origins of paternally derived OS. The origins of OS that could affect fathers and their gametes mainly derive from four causes: male fertility complications, lifestyle/diet, environmental/occupational exposures, and special treatments. OS, oxidative stress; MAGI, male accessory gland infections; MOSI, male oxidative stress infertility.

**Figure 2 ijms-21-05175-f002:**
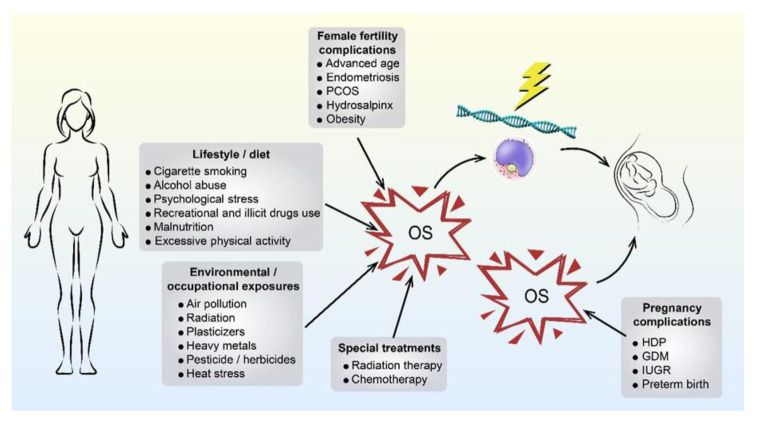
This figure shows the origins of maternally derived OS. The origins of OS that could affect the mothers, gametes, and their pregnancies mainly derive from five aspects: female fertility complications, lifestyle/diet, environmental/occupational exposures, special treatments and pregnancy complications. PCOS, polycystic ovary syndrome; HDP, hypertensive disorders of pregnancy; GDM, gestational diabetes mellitus; IUGR, intrauterine growth restriction.

**Figure 3 ijms-21-05175-f003:**
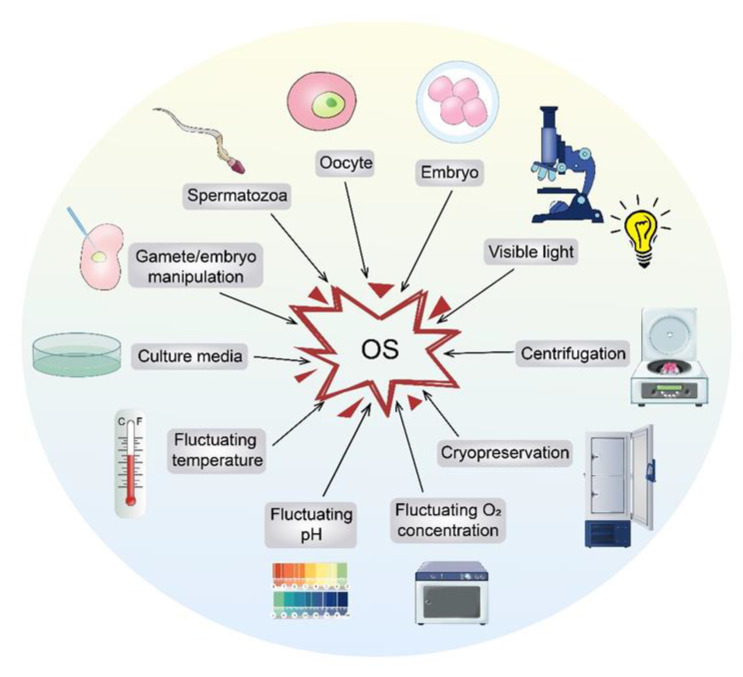
This figure shows the origins of ART-derived OS. During ART, several factors might lead to elevated OS. ART, assisted reproductive technology.

**Figure 4 ijms-21-05175-f004:**
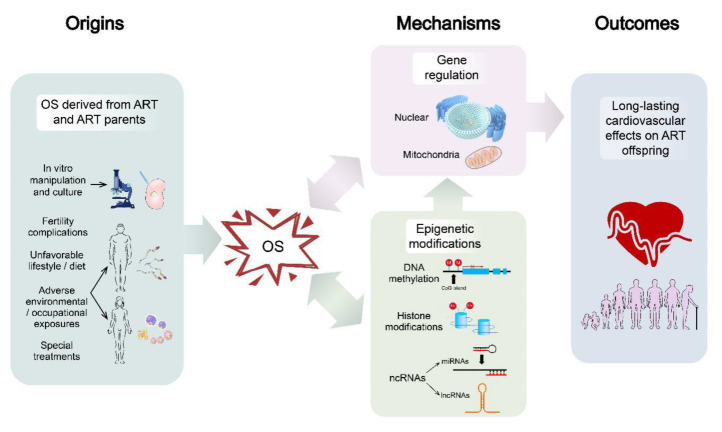
This figure shows the early-life OS exert long-lasting effects on offspring conceived by ART. ncRNAs, non-coding RNAs; miRNAs, micro-RNAs; lncRNAs, long non-coding RNAs.

**Table 1 ijms-21-05175-t001:** Antioxidant genes regulated by Nrf2.

Gene	Protein Encoded	Synonyms	Species ^1^	Refs
**GSH-based antioxidant genes**				
*GCLC*	Glutamate-cysteine ligase catalytic subunit	*GCS, GLCL, GLCLC*	m, h	[131,132,133]
*GCLM*	Glutamate-cysteine ligase modifier subunit	*GLCLR*	m, h	[131,132,133,134]
*GGT1*	Gamma-glutamyltransferase 1	*CD224, D22S672, D22S732, GGT*	h	[131]
*GLRX*	Glutaredoxin	*GRX, GRX1*	h	[131]
*GLS*	Glutaminase	*GLS1, KIAA0838*	h	[131]
*GPX1*	Glutathione peroxidase 1	-	m	[135,136]
*GPX2*	Glutathione peroxidase 2	*GSHPX-GI*	m, h	[131,137,138,139,140]
*GPX4*	Glutathione peroxidase 4	*MCSP, PHGPx*	m	[131]
*GSR*	Glutathione-disulfide reductase	-	m, h	[133,134]
*GSTA1*	Glutathione S-transferase alpha 1	-	m	[131,141,142,143]
*GSTA2*	Glutathione S-transferase alpha 2	-	m	[132,142,143]
*GSTA3*	Glutathione S-transferase alpha 3	-	m	[141,142,143]
*GSTA4*	Glutathione S-transferase alpha 4	-	m	[144]
*GSTM1*	Glutathione S-transferase mu 1	*GST1, H-B, MU*	m	[131,132,143,144]
*GSTM2*	Glutathione S-transferase mu 2	*GST4*	m	[132,143,144]
*GSTM3*	Glutathione S-transferase mu 3	*GST5*	m, h	[132,134,143,144]
*GSTM4*	Glutathione S-transferase mu 4	-	m	[144]
*GSTM5*	Glutathione S-transferase mu 5	-	m	[145]
*GSTM6*	Glutathione S-transferase mu 6	-	m	[144]
*GSTP1*	Glutathione S-transferase pi 1	*FAEES3, GST3, GSTP*	m	[131]
*MGST1*	icrosomal glutathione S-transferase 1	*GST12, MGST-I*	m, h	[131]
*MGST2*	Microsomal glutathione S-transferase 2	*MGST-II*	m	[143]
*MGST3*	Microsomal glutathione S-transferase 3	*GST-III*	m	[132,143]
*SLC6A9*	Solute carrier family 6 member 9	*GLYT1*	m	[131]
*SLC7A11*	Solute carrier family 7 member 11	*xCT*	m, h	[131,146]
**TXN-based antioxidant genes**				
*PRDX1*	Peroxiredoxin 1	*NKEFA, PAGA)*	m	[131,138,142]
*PRDX6*	Peroxiredoxin 6	*1-Cys, aiPLA2, AOP2, KIAA0106, MGC46173, NSGPx, p29, PRX*	h	[131,147]
*SRXN1*	Sulfiredoxin 1	*C20orf139, dJ850E9.2, Npn3, SRX1, YKL086W*	m, h	[131]
*TXN*	Thioredoxin	*TRX*	m, h	[131,132,135]
*TXNRD1*	Thioredoxin reductase 1	*GRIM-12, Trxr1, TXNR*	m, h	[131,135,144]
**ATP-binding-based antioxidant genes**				
*ABCB6*	ATP binding cassette subfamily B member 6	*EST45597, MTABC3, umat*	m, h	[131]
*ABCC1*	ATP binding cassette subfamily C member 1	*GS-X, MRP, MRP1*	m, h	[131]
*ABCC2*	ATP binding cassette subfamily C member 2	*CMOAT, cMRP, DJS, MRP2*	m, h	[131,141]
*ABCC3*	ATP binding cassette subfamily C member 3	*cMOAT2, EST90757, MLP2, MOAT-D, MRP3*	m, h	[131,141,148]
*ABCC4*	ATP binding cassette subfamily C member 4	*EST170205, MOAT-B, MOATB, MRP4*	m	[131]
*ABCC5*	ATP binding cassette subfamily C member 5	*EST277145, MOAT-C, MRP5, SMRP*	m	[131]
**Heme/iron metabolism-associated antioxidant genes**				
*BLVRA*	Biliverdin reductase A	*BLVR*	h	[131]
*BLVRB*	Biliverdin reductase B	*FLR, SDR43U1*	m, h	[131]
*FTH1*	Ferritin heavy chain 1	*FHC, FTH, FTHL6, PIG15, PLIF*	m, h	[131]
*FTL*	Ferritin light chain	*MGC71996, NBIA3*	m, h	[131]
*HMOX1*	Heme oxygenase 1	*bK286B10, HO-1*	m, h	[131,133,134,135,140,145]
**UDP glucuronosyltransferase-associated antioxidant genes**				
*UGT1A1*	UDP glucuronosyltransferase family 1 member A1	*GNT1, UGT1, UGT1A*	h	[131]
*UGT1A6*	UDP glucuronosyltransferase family 1 member A6	*GNT1, HLUGP, UGT1F*	m	[138]
*UGT2B1*	UDP glucuronosyltransferase family 2 member B1	-	m	[141]
*UGT2B5*	UDP glucuronosyltransferase family 2 member B5	-	m	[132,143]
*UGT2B7*	UDP glucuronosyltransferase family 2 member B7	*UGT2B9*	m, h	[131]
**Other antioxidant genes**				
*ADH7*	Alcohol dehydrogenase 7 (class IV), mu or sigma polypeptide	*ADH-4*	m	[131]
*AKR1A1*	Aldo-keto reductase family 1 member A1	*ALR, DD3*	h	[132,143]
*AKR1B1*	Aldo-keto reductase family 1 member B1	*ALDR1, AR*	m, h	[131]
*AKR1B8*	Aldo-keto reductase family 1 member B8	-	m	[142,143]
*AKR1C1*	Aldo-keto reductase family 1 member C1	*DD1, DDH, DDH1, HAKRC, MBAB*	h	[131]
*ALDH1A1*	ldehyde dehydrogenase 1 family member A1	*ALDH1, PUMB1, RALDH1*	m	[131]
*ALDH3A1*	Aldehyde dehydrogenase 3 family member A1	*ALDH3*	m, h	[131]
*ALDH7A1*	Aldehyde dehydrogenase 7 family member A1	*ATQ1, EPD, PDE*	m	[131]
*CAT*	Catalase	-	m	[137,141]
*CBR1*	Carbonyl reductase 1	*CBR, SDR21C1*	h	[131]
*CYP1B1*	Cytochrome P450 family 1 subfamily B member 1	*CP1B, GLC3A*	m	[131]
*CYP2B9*	Cytochrome P450 family 2 subfamily B member 9	-	m	[131]
*G6PD*	Glucose-6-phosphate dehydrogenase	*G6PD1*	m, h	[131]
*IDH1*	Isocitrate dehydrogenase (NADP(+)) 1, cytosolic	-	m	[131]
*ME1*	Malic enzyme 1	-	m, h	[131]
*NQO1*	NAD(P)H quinone dehydrogenase 1	*DHQU, DIA4, DTD, NMOR1, QR1*	m, h	[131,132,134,135,142,143]
*PGD*	Phosphogluconate dehydrogenase	-	m, h	[131]
*PTGR1*	Prostaglandin reductase 1	*LTB4DH, ZADH3*	h	[131]
*SOD1*	Superoxide dismutase 1	*ALS, ALS1, IPOA*	m	[141]
*SOD2*	Superoxide dismutase 2	-	m	[141]
*SOD3*	Superoxide dismutase 3	*EC-SOD*	m	[142]
*TALDO1*	Transaldolase 1	-	m, h	[131]
*UGDH*	UDP-glucose 6-dehydrogenase	-	h	[131]

^1^ The species means the gene has been identified in mouse (m) and/or human (h). Nrf2, nuclear factor erythroid 2-related factor; GSH, glutathione; UDP, uridine diphosphate. The GSH and TXN antioxidant pathways are two important downstream pathways of Nrf2 [117].

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
