# Peer review of "Early Life Oxidative Stress and Long-Lasting Cardiovascular Effects on Offspring Conceived by Assisted Reproductive Technologies: A Review"

_ijms, 2020, doi:10.3390/ijms21155175_

Round 1

Reviewer 1 Report

Interesting concept in an attempt to explain known adverse outcomes of ART. Despite the lack of robust evidence it is definitely worth publishing. I think the conclusion should state some more uncertainty regarding the relation of OS, infertility and long term ART adverse outcomes.

Author Response

To Reviewer 1:

Interesting concept in an attempt to explain known adverse outcomes of ART. Despite the lack of robust evidence it is definitely worth publishing. I think the conclusion should state some more uncertainty regarding the relation of OS, infertility and long term ART adverse outcomes.

Answer: Thank you very much for your approval and your valuable comment! Hope this paper could provide some enlightenments for future research in this field. We redraw a more cautious conclusion on page 11, lines 328-330.

Reviewer 2 Report

The paper by Huixia Yang et al.

“Early Life Oxidative Stress and Long-lasting 2 Cardiovascular Effects on Offspring Conceived by 3 Assisted Reproductive Technologies: A Review”,

the origins of OS, increased after Assisted reproductive technology (ART), the long-lasting cardiovascular effects and potential mechanisms of OS on the produced offspring are discussed.

The manuscript is original and interest in its field.

However, there are some minor revisions before being accepted:

Minore revision:

  • The authors should improve the english language.

  • In the paragraph the Origins of OS, the author should insert a part with the traditional treatments used against OS, focusing on the natural compounds.

     For example: DOI:10.3390/nu11092175

  • The author should improve the bibliography:

In the paragraph "Nrf2-Mediated Anti-OS Signaling Pathway" the author should insert a part related the pre clinical studies on cardiovascular complications treated with antioxidant compounds:

       For example: DOI:10.1016/j.nbd.2019.01.007

  • The author should insert in the part related to the cardiovascular complications several pre clinical studies, where the antioxidant compounds contrast the hypoxia / reperfusion damage.

       For example: DOI: 10.3390/ijms20194845

Author Response

To Reviewer 2:

The authors should improve the english language.

Answer: This paper has been proofread by Rhine Language Editing Limited. Attached please find the certificate of language editing for this paper.

In the paragraph the Origins of OS, the author should insert a part with the traditional treatments used against OS, focusing on the natural compounds.

Answer: Thank you very much for the valuable comments and the reference papers, both of them provided us great inspirations for completing a more comprehensive review. As recommended, we have insert a part (page 5, lines 156-164) in the paragraph “Origins of OS”.

The author should improve the bibliography: In the paragraph "Nrf2-Mediated Anti-OS Signaling Pathway" the author should insert a part related the pre clinical studies on cardiovascular complications treated with antioxidant compounds:

Answer: Related studies have been added (page 7, lines 230-232) in the paragraph "Nrf2-Mediated Anti-OS Signaling Pathway" in the revised version.

The author should insert in the part related to the cardiovascular complications several pre clinical studies, where the antioxidant compounds contrast the hypoxia / reperfusion damage.

Answer: Related studies have been added (page 7, lines 232-234) in the paragraph "Nrf2-Mediated Anti-OS Signaling Pathway" in the revised version.
